# Fucoid Macroalgae Have Distinct Physiological Mechanisms to Face Emersion and Submersion Periods in Their Southern Limit of Distribution

**DOI:** 10.3390/plants10091892

**Published:** 2021-09-14

**Authors:** Maria Martins, Cristiano Soares, Inês Figueiredo, Bruno Sousa, Ana Catarina Torres, Isabel Sousa-Pinto, Puri Veiga, Marcos Rubal, Fernanda Fidalgo

**Affiliations:** 1GreenUPorto—Sustainable Agrifood Production Research Centre and INOV4AGRO, Biology Department, Faculty of Sciences University of Porto (FCUP), Rua do Campo Alegre, 4149-007 Porto, Portugal; bruno.filipe@fc.up.pt (B.S.); ffidalgo@fc.up.pt (F.F.); 2Biology Department, Faculty of Sciences University of Porto (FCUP), Rua do Campo Alegre, 4149-007 Porto, Portugal; c.fsoares@fc.up.pt (C.S.); up201505847@fc.up.pt (I.F.); 3Interdisciplinary Centre of Marine and Environmental Research (CIIMAR), University of Porto, Novo Edifício do Terminal de Cruzeiros do Porto de Leixões, Avenida General Norton de Matos, 4450-208 Matosinhos, Portugal; a_catarina_torres@hotmail.com (A.C.T.); ispinto@fc.up.pt (I.S.-P.); puri.sanchez@fc.up.pt (P.V.); marcos.garcia@fc.up.pt (M.R.)

**Keywords:** *Pelvetia canaliculata*, *Ascophyllum nodosum*, *Fucus serratus*, intertidal macroalgae, oxidative stress

## Abstract

During high tide, macroalgae are submersed, facing adequate environmental conditions, however, at low tide, these species can be exposed to high UV radiation and desiccation, leading to an overproduction of reactive oxygen species, causing oxidative stress. Since intertidal organisms present differential sensitivity to abiotic fluctuations, this study aimed to evaluate the physiological responses [photosynthetic pigments, hydrogen peroxide (H_2_O_2_), lipid peroxidation (LP), and thiols and proline] of three macroalgae, from different intertidal levels, towards tidal regimes. Samples of *Pelvetia canaliculata*, *Ascophyllum nodosum*, and *Fucus serratus* were collected from beaches located on the southern limit of distribution in periods of potential stress (Summer and Spring), under low and high tide. The photosynthetic pigments of *P. canaliculata* and *F. serratus* were generally higher during low tide, and the oxidative damage evidenced by H_2_O_2_ and LP increased in the Summer, while *A. nodosum* showed greater oxidative damage in the Spring. While thiol content did not change, proline levels were species- and tidal-specific among sampling dates. *P. canaliculata* presented higher resilience to unfavorable conditions, while *F. serratus* was the most sensitive species. The physiological responses analyzed were species-specific, pointing to the high susceptibility of low intertidal organisms to expected extreme climatic events.

## 1. Introduction

Climate change can exert a significant role in coastal and marine ecosystems, affecting their structure and functioning. In coastal zones, macroalgae, as photosynthetic organisms, are one of the bases of marine food webs, essential for maintaining their regular functioning [1]. Most macroalgae species live in the intertidal zone of a rocky shore [2], a zone characterized by its high environmental instability, due to the accentuated fluctuations of environmental factors (e.g., temperature, salinity, desiccation, radiation) which will be exacerbated given the current climate change scenario. Thus, the capacity of these species to withstand the environmental challenges determines their distribution and abundance along the intertidal zone, allowing the organisms to survive and reproduce [2]. Moreover, understanding the physiological plasticity of intertidal seaweeds towards an ever-changing environment, where differences between tidal regimes are increasingly challenging, can provide important clues about how climate change will impact the distribution pattern of these organisms.

Intertidal macroalgae, as sessile organisms, are periodically subjected to UV radiation and high temperatures, due to desiccation during emersion, followed by submersion, where they face homeostasis-promoting conditions. As a consequence of extended desiccation, the efficiency of the photosystems I and II is disturbed, thus negatively affecting the photosynthetic rate [3]. These physiological disturbances disrupt the redox homeostasis by an overproduction of reactive oxygen species (ROS), that comprise different chemical species, including molecular [hydrogen peroxide (H_2_O_2_) and singlet oxygen (^1^O_2_)] and radical [superoxide anion (O_2_^.-^) and hydroxyl radical (OH^.^)] forms [2,4]. Although ROS are constantly produced as a result of several metabolic processes, and at low concentrations, they are recognized as important signaling compounds, at high levels, they can be very harmful to the cells inducing the occurrence of oxidative stress. However, to prevent oxidative bursts, macroalgae developed a tightly regulated antioxidant (AOX) system, that includes a variety of enzymatic (e.g., superoxide dismutase—SOD, EC 1.15.1.1; catalase—CAT, EC 1.11.1.6; ascorbate peroxidase—APX, EC 1.11.1.11) and non-enzymatic (e.g., ascorbic acid—AsA; glutathione—GSH; proline) components [2,4]. Acting synergistically with each other, AOX enzymes and metabolites can reduce the harmful effects of ROS overproduction and accumulation, reestablishing redox homeostasis. Nonetheless, as a result of the severe and/or repeated stress events, there can be either an overproduction of ROS and/or an inhibition of the AOX defense, leading to ROS levels that exceed the AOX capacity of these organisms. As a consequence, lipid peroxidation (LP) can occur, causing damage to the membranes and the photosynthetic apparatus, in addition to promoting the degradation of nucleic acids and proteins [2,4].

*Pelvetia canaliculata* (L.) Decaisne and Thuret, *Ascophyllum nodosum* (L.) Le Jolis and *Fucus serratus* L. are three brown macroalgae species belonging to the Fucales family that inhabit the rocky intertidal shores and are widely distributed around the North Atlantic [5,6]. *Pelvetia canaliculata*, usually found at the upper limit of the intertidal [7], is known as one of the most stress-tolerant macroalgae species, as it can resist desiccation and rapidly acclimates to high radiation, mainly due to its carotenoid composition [7]. *Ascophyllum nodosum* and *Fucus serratus*, living in the mid-high and mid-low intertidal zone, respectively, play an important role in coastal ecosystems, as they provide food, habitat, and shelter for a variety of living beings, along with the supply of essential nutrients [5,6]. In addition, these two species, despite being ecologically and phylogenetically close, differ in their life-spans and growth rates, which may evidence different capacities for adaption and response to environmental constraints [5]. As the Iberian Peninsula is the southern distributional limit for these cold-water species and given that in recent decades this area has been largely affected by climate change [8], the abundance of these organisms in this area has been declining [9]. Thus, efforts need to be made to prevent future adaptations and/or shifts in the distribution of intertidal living beings. Up to date, the scientific community has been dedicated to unraveling the response of individual macroalgae species living in the intertidal and, in most studies, under controlled conditions [3,10,11,12]. However, research should be conducted under field conditions, i.e., a more realistic scenario, to better understand the physiological responses of macroalgae that live at distinct intertidal levels to the above-mentioned environmental fluctuations to unravel the adaptation mechanisms that these species employ to deal with the ever-changing conditions that they are subjected to.

Bearing in mind the influence of multiplicity of environmental constraints and the complexity of their interactions in the growth and development of *P. canaliculata*, *A. nodosum*, and *F. serratus*, the present study aimed to unravel the impact of the tidal regime on the physiological performance of these macroalgae species under field conditions. The main hypothesis that supports this work was that the three selected macroalgae species, depending on their position in the intertidal zone, possess distinct physiological mechanisms to overcome desiccation and hydration cycles. For this purpose, three samplings were carried out between August 2017 and August 2018, corresponding to potential periods of stress, during submersion and emersion periods, and several biochemical endpoints (photosynthetic pigments, H_2_O_2_, proline, LP, and thiols) were evaluated.

## 2. Results

### 2.1. Photosynthetic Pigments

Chlorophylls and carotenoids levels showed significant differences for the Species × Date interaction (Sp × Da; Appendix A), indicating that the content of photosynthetic pigments of the three species studied was variable among the dates. Moreover, significant differences were detected between the two levels of the tide factor (low and high tide), regardless of species and date.

*Fucus serratus* exhibited the highest chlorophyll amount (Figure 1a) at low and high tides in August 2017, being 60% and 100% higher than those observed for *P. canaliculata* and *A. nodosum*, respectively. In the last sampling date (August 2018), the levels of chlorophyll of *F. serratus* and *P. canaliculata* were 130% and 60% higher at low tide than those of *A. nodosum*, respectively. Regarding differences between tides, the content of these pigments of *P. canaliculata* was significantly higher during low tide (up to 60%) in all sampling dates, when compared to the high tide. On the other hand, *F. serratus* showed a more variable pattern in the amount of chlorophyll at low tide, with rises observed at low tide in August 2017 (50%) and August 2018 (100%). *Ascophyllum nodosum* showed no differences between tides.

Regarding the carotenoids content (Figure 1b), in August 2017 and March 2018, the three species showed differences between them in low tide conditions, with *A. nodosum* having significantly lower levels than the remaining species in both sampling dates. Moreover, in August 2017, *F. serratus* had the highest levels of carotenoids, while *P. canaliculata* was the dominant species in March 2018. In terms of differences between low and high tides in the same species, an increase in the content of carotenoids at low tide was found in *P. canaliculata* in March (50%) and August 2018 (30%). The same pattern was also observed for *F. serratus* in August 2017 and August 2018, with rises at low tide conditions up to 70%. No differences between tides were found for *A. nodosum*.

### 2.2. Redox Status

Regarding H_2_O_2_ content, significant differences were recorded for the Species × Date interaction (Sp × Da; Appendix A), indicating that the three species had variable levels of this ROS among the sampling dates.

In addition, since no significant differences between tides were detected, the results were presented as the average of both tides for each species and date. As shown in Figure 2a, H_2_O_2_ levels showed statistical differences between species in the three sampling dates: while *P. canaliculata* had the highest H_2_O_2_ content on both Summer dates (August 2017 and 2018), the same pattern was not observed in the Spring, where *A. nodosum* exhibited the highest amount of this ROS. Moreover, in August 2018, there were no differences in the H_2_O_2_ levels of *F. serratus* and *A. nodosum*.

Concerning total thiols, significant differences for the Species × Tide interaction (Sp × Ti; Appendix A) were found.

Since no differences between sampling dates were detected, the results were presented as the average of each date, for the respective tidal regime, for each species. As can be seen, under a high tide, no differences between species were recorded (Figure 2c). However, at low tide, *P. canaliculata* and *F. serratus* presented the highest thiols levels, being 52% and 73% higher than those of *A. nodosum*, respectively.

The degree of LP was evaluated by quantifying malondialdehyde (MDA). The ANOVA analyses revealed significant differences regarding MDA levels for the Species × Tide × Date interaction (Sp × Ti × Da; Appendix A). As can be seen in Figure 2b, *P. canaliculata* and *F. serratus* had the highest MDA content in August 2017 and 2018, respectively, in the two tidal regimes. In March 2018 (date 2), MDA levels in *A. nodosum* were significantly increased when compared to those quantified for *P. canaliculata* and *F. serratus* at low tide.

Regarding tidal differences, on the first sampling date, *P. canaliculata* at low tide had significantly higher (60%) MDA levels than at high tide, remaining similar between tides in March and August 2018. *Ascophyllum nodosum* showed an increase (100%) of MDA under a low tide when compared to the high tide in March 2018. No difference between tides was found for *F. serratus* on all sampling dates.

Regarding proline content, ANOVA analyses revealed significant differences for the Species × Tide × Date interaction (Sp × Ti × Da; Appendix A).

Post-hoc analyses showed that in the first two sampling dates (August 2017 and March 2018), *P. canaliculata* had the lowest levels of proline (Figure 2d), followed by *A. nodosum* and *F. serratus*, while the opposite was found in date 3, where *P. canaliculata* had the highest levels of proline. Regarding tide effects within the same species, only *A. nodosum* and *F. serratus* displayed statistical differences between high and low tides: on dates 1 and 3, the proline content of *A. nodosum* decreased and increased, respectively, 60% at low tide; on date 2, the levels of this amino acid at low tide increased 30% in *F. serratus*.

## 3. Discussion

As a result of climate change, intertidal macroalgae will face an even more challenging habitat with extreme environmental conditions [13]. Influenced by this climatic instability, cold-water macroalgae are shifting their distribution patterns along the coastline, thus reducing their abundance in the southern limit areas of the Iberian Peninsula [9]. Therefore, it is of particular relevance to understanding the physiological responses of seaweeds to unfavorable conditions, in order to predict their adaptability in a future marked by climate change. Regarding this matter, although several laboratory studies have been conducted to assess the impact of single abiotic stressors (e.g., temperature, salinity, desiccation) on these organisms [3,13,14], the physiological responses of macroalgae under field conditions can be quite distinct, thus highlighting the importance of these types of studies [12,15]. Thus, this work aimed to evaluate how three species (*P. canaliculata, A. nodosum*, and *F. serratus*) belonging to distinct intertidal levels, modulated their physiological processes between high and low tide periods, particularly exploring their redox status.

Changes in solar irradiance throughout the day and different tidal levels can influence the total amount of light that reaches intertidal species, making this a limiting and variable factor for these organisms [13]. Therefore, to regulate the number of photons absorbed by their photosynthetic apparatus, macroalgae have developed some photoprotective strategies, including the adjustment of the chloroplast orientation and repair mechanisms, activation of the xanthophyll cycle by carotenoids, and, particularly in the case of brown seaweeds, the accumulation of UV-absorbing phenolic compounds [16]. In this way, photosynthetically active cells can optimize the use of light, when organisms face conditions in which the amount of this factor compromises photosynthesis [13]. Our results showed that the content of photosynthetic pigments of all species increased during low tide, in some cases significantly, over the sampling dates. During long emersion periods, macroalgae suffer water loss, which strongly affects their photosynthetic rate due to the oxidation of pigments and proteins [3]. Despite this, the data herein collected points to an opposite trend. Yet, it should be highlighted that, during short-term desiccation, as was the case in this study, an increase in the levels of photosynthetic pigments can be observed [17]. In addition, the increase in carotenoids content may be related to its AOX capacity (in particular β-carotene) in the direct elimination of ^1^O_2_, by quenching excited chlorophyll or dissipating excess energy via the xanthophyll cycle [13,15]. Furthermore, it was noticed that in almost all sampling dates, *P. canaliculata* had the highest content of carotenoids, followed by *F. serratus* and *A. nodosum*, regardless of the tidal regime. The high concentration of carotenoids in *P. canaliculata* has been previously observed [14], being pointed out as a photoprotective defense against UV radiation. Additionally, this species is known to promptly activate non-photochemical quenching mechanisms, such as the xanthophyll cycle, in order to reduce excess energy, converting violaxanthin into antheraxanthin and zeaxanthin, an important response mechanism to high light intensity [13]. On the other hand, the absence of changes in *A. nodosum* carotenoid levels between tides in all sampling dates may indicate that these pigments did not play a photoprotective role in this species, explaining the occurrence of oxidative stress.

Due to the instability of their habitat, the metabolic functions of intertidal macroalgae are constantly altered, increasing ROS production, which, in turn, can lead to oxidative stress [3,12].

In the present study, the production of H_2_O_2_, an important ROS, the degree of LP, an oxidative stress biomarker, and the thiol content, indicative of cellular redox homeostasis, were evaluated. As proline has an important AOX and osmotic role, this metabolite was also quantified. Regarding H_2_O_2_ levels, no differences were found between the tides for the three species on all sampling dates, indicating that this ROS is not the best parameter to assess responses related to the tidal level. Contrary to these results, previous laboratory studies reported that under desiccation-induced stress, macroalgae had a higher H_2_O_2_ content when compared to the period of submersion [3,17]. The discrepancy between these results and those obtained in the present study may be precisely related to the conditions tested, since, in our study, the macroalgae were collected in their native environment, where the organisms were subjected to a combination of abiotic and biotic factors, which may lead to different responses from those obtained in laboratory experiments [12,15]. Despite the absence of differences between tides, the results were species-specific. *F. serratus* and *P. canaliculata* clearly presented higher levels of this ROS in August 2017 and 2018, corresponding to the two potential stress periods, with the maximum H_2_O_2_ content of all tested species achieved by *P. canaliculata*, although with no apparent change between tides, suggesting a great production of basal levels of ROS in this species. On the other hand, the levels of H_2_O_2_ in *A. nodosum* were the highest in March 2018. Due to the Mehler reaction, in which the O_2_ is reduced to H_2_O_2_, large amounts of this ROS are produced during photosynthesis [4]. Thus, the high H_2_O_2_ content of *A. nodosum* in the above-mentioned sampling dates pairs with the increase in chlorophyll content, suggesting a higher photosynthetic rate during that period of the year. Additionally, the H_2_O_2_ concentration quantified in this species were corroborated by those found for the MDA content, the latter showing the same pattern of variation, since there was a significant increase in the degree of LP at low tide in March 2018, that was accompanied by a small decrease in Pro levels. However, the redox homeostasis of the cell, which was evaluated by the total thiols, did not appear to be affected during low tide. Gathering all the results of *A. nodosum*, it might be suggested that this species suffered oxidative damage in March 2018 that was aggravated during low tide. Yet, H_2_O_2_ might not have been the ROS responsible for the rise of LP degree, which may support the role of H_2_O_2_ as a signaling molecule during photosynthesis [18]. The same response also seemed to occur in *P. canaliculata* in August 2017, as the LP degree significantly increased at low tide, despite the absence of differences in H_2_O_2_. It is known that *P. canaliculata*, by living on the upper shore of the intertidal, often faces several unfavorable conditions [13]. Thus, the oxidative damage of *P. canaliculata* was expected to be higher, when compared to the other two studied species. In accordance, our results showed that *P. canaliculata* presented the highest levels of H_2_O_2_, thiols, and MDA, regardless of the tidal regime, thus evidencing not only the harsh environment in which this species lives but also suggesting its high resilience [13]. Despite the physiological mechanisms of this species to withstand the extreme conditions still being unknown, some authors suggest the AOX role of carotenoids, corroborating the observed significantly higher levels of these pigments under low tide, and phlorotannins [14]. Regarding *F. serratus*, the species that live on the lowest shore of the intertidal, its oxidative damage, evidenced by H_2_O_2_, MDA, and thiols content, was often higher than *A. nodosum* during the low tide of August 2017 and 2018, demonstrating the high sensitivity of *F. serratus*. In fact, it was suggested that algae species living at the low intertidal cannot overcome the excess of ROS during low tide, while the ones that live in the top limits of the intertidal possess mechanisms that successfully minimize the oxidative damage imposed by their habitat [3]. Furthermore, *F. serratus* presented similar or higher levels of some parameters (e.g., thiols and MDA content) than those of *P. canaliculata*, demonstrating the vulnerability of *F. serratus* to variability between tides. A previous study, which evaluated the influence of the vertical distribution of macroalgae in the intertidal on stress resilience of three species (*F. distichus*, *F. spiralis*, *F. evanescens*), found similar levels of MDA between *F. distichus*, found at the lowest intertidal level, and *F. spiralis*, which occupied a higher position on the shore when exposed to freezing and desiccation-induced stress [10]. Paired with these results, the present study supported the evidence that macroalgae species that face low periods of emergence are more susceptible to unfavorable conditions, reaching similar levels of oxidative damage to species that are almost permanently exposed to several abiotic stressors.

Although thiols content of *F. serratus* and *P. canaliculata* were significantly higher than *A. nodosum*, there were no differences between tides, although the observed tendency for increased levels of these compounds during low tide, mainly in *P. canaliculata*. These results may suggest the role of some thiol-containing compounds, such as GSH, in the response of these algae species to their unstable habitat. Proline levels of *P. canaliculata* showed no differences between tides in the three sampling dates, suggesting that proline appears not to have been involved in the AOX response of this species. The proline content in *F. serratus* increased at low tide, particularly in March 2018, supporting the osmoprotective role of this amino acid and its action as a membrane stabilizer, preventing the increase of LP [4].

## 4. Materials and Methods

### 4.1. Sampling Procedure and Experimental Design

Individuals of *P. canaliculata, A. nodosum*, and *F. serratus* were collected in Viana do Castelo, Portugal (between 41°43′0.3′′ N and 41°41′36.36′′ N; 8°51′10.52′′ W), since it is the only shore where these three species coexist at their southern limit of distribution. Sampling was conducted on three distinct dates: August 2017, March 2018, and August 2018, under low and high tidal regimes.

All the organisms were collected at the middle of the high tide period, corresponding to a time where species were in their optimal growth conditions, and at the end of the low tide, right before the water reached the respective intertidal level of each species. During the high tide period, water depth was 0.5–0.75 m for *P. canaliculata*, 1.0–1.30 m for *A.nodosum*, and *F. serratus*, the species belonging to the lowest intertidal level, covered by 2.0–3.0 m seawater. Furthermore, at low tide, sampling was conducted immediately before the water covered the species to achieve the maximum exposure period for each studied macroalgae species. Considering that throughout a tidal cycle, which has a 12 h period and corresponds to low and high tide periods, the longest emersion time was recorded for *P. canaliculata* at 10 h, while *A. nodosum* and *F. serratus* were exposed during 8 h and 4 h, respectively. Since the main goal was to verify how macroalgae living in distinct intertidal levels responded to different tidal regimes, the chosen dates corresponded to potential periods of stress (two during summer and one during spring), as these macroalgae are known as cold-water species. Information about water temperature, atmospheric temperature, and UVB irradiance, which was calculated through UV index (http://www.ipma.pt/pt/oclima/monitoriza.dia/; accessed on 20 July 2021) and according to the formula [UVB_280–315 nm_ (µW cm^−2^) = 7.55 × UVEry (correspondent to erythemal weighted UV radiation)] described by McKenzie et al. [19], for each sampling date is summarized in Table 1.

Since this work was conducted under field conditions, to minimize the effects of intrinsic variability, for each date and tidal period, five biological replicates of each macroalgae species were collected. Immediately after cutting, macroalgae were frozen in liquid nitrogen. Upon arrival to the laboratory, the plant material was ground into liquid nitrogen and stored at −80 °C for subsequent biochemical procedures.

### 4.2. Biochemical Endpoints

#### 4.2.1. Extraction and Quantification of Photosynthetic Pigments

The content of total chlorophylls and carotenoids was quantified according to Lichtenthaler [20]. Frozen samples were homogenized in 80% (*v*/*v*) acetone and centrifuged at 1400× *g* for 10 min. Afterward, the absorbance at 470, 647, and 663 nm was recorded and the chlorophylls and carotenoids levels were calculated using the formulas of Lichtenthaler [20]. Results were expressed as mg g^−1^ fresh weight (f.w.).

#### 4.2.2. Hydrogen Peroxide (H_2_O_2_) Determination

H_2_O_2_ quantification was performed following the procedures of Jana and Choudhury [21]. Samples were homogenized on ice in 50 mM potassium phosphate buffer (pH 6.5) and centrifuged at 6000× *g* for 25 min. Then, the supernatant was mixed with 0.1% (*v*/*v*) TiSO_4_ in 20% (*v*/*v*) H_2_SO_4_, and samples were vigorously vortexed and centrifuged in the same conditions as previously described. Finally, the absorbance at 410 nm was measured and the H_2_O_2_ content was calculated using the extinction coefficient of 0.28 µM^−1^ cm^−1^. The results were expressed in nmol g^−1^ f.w.

#### 4.2.3. Evaluation of Lipid Peroxidation (LP)

LP was evaluated through the measurement of malondialdehyde (MDA) content by the method of Heath and Packer [22]. After homogenization of the samples in 0.1% (*w*/*v*) trichloroacetic acid (TCA) and centrifugation (10,000× *g*; 5 min), the supernatant reacted with 0.5% thiobarbituric acid in 20% (*w*/*v*) TCA for 30 min at 95 °C. Tubes were then cooled and centrifuged (10,000× g; 7 min), and the absorbances at 532 and 600 nm were recorded. To minimize the unspecific turbidity, the obtained values at 600 nm were subtracted to the ones of 532 nm. Finally, the levels of MDA were estimated through the extinction coefficient of 155 mM^−1^ cm^−1^ and expressed as nmol^−1^ f.w.

#### 4.2.4. Quantification of Total Thiols

The spectrophotometrical quantification of thiols was achieved following the procedure described by Zhang et al. [23]. Samples were homogenized, on ice, in a solution of 20 mM EDTA and 20 mM ascorbic acid and centrifuged at 13,000× *g* for 20 min at 4 °C. Then, 200 mM Tris-HCl (pH 8.2) and 10 mM 5,5’-dithiobis (2-nitrobenzoic acid) was added to the supernatant. Mixtures were incubated for 15 min at room temperature and the absorbance was then registered at 412 nm. The extinction coefficient of 13,600 M^−1^ cm^−1^ was used to calculate the levels of total thiols, being the results expressed as nmol g^−1^ f.w.

#### 4.2.5. Proline Quantification

Proline quantification was conducted according to Bates et al. [24], using the ninhydrin colorimetric assay. Macroalgae samples were homogenized in 3% (*w*/*v*) sulphosalicylic acid and centrifuged at 500 × *g* for 10 min, followed by an incubation period of 1 h at 95 °C, in which the supernatant was mixed with glacial acetic acid and ninhydrin. Finally, toluene was added to extract the proline-ninhydrin complex and the absorbance was recorded at 520 nm. The levels of proline were calculated using a calibration curve, previously prepared with known concentrations of this amino acid, and results were expressed in mg g^−1^ f.w.

### 4.3. Statistical Analysis

For each species, five individuals were collected on each date and for each tide regime. All the laboratory procedures were conducted, at least, in triplicate for each replicate to control the quality of the procedure. Results are expressed as mean ± standard deviation (STDEV) of the five replicates. To explore the differences between the considered factors, a three-way ANOVA was conducted, being the species (*A. nodosum*, *F. serratus*, and *P. canaliculata*) and the tide (low and high) defined as fixed factors, and the date as a random factor. Whenever significant (*p* < 0.05), the ANOVAs were followed by the posthoc test of Student-Newman-Keuls (SNK).

## 5. Conclusions

In a climate change-related scenario, understanding intertidal macroalgae responses towards environmental disturbances in field conditions is of utmost importance, in order to predict future distributional patterns of these species along the coastline. Thus, the results of this work strengthen that macroalgae that live at different levels of the intertidal show species-signature physiological responses, being able to modulate their stress responses to the environmental conditions found between high and low tides. Indeed, in the present study, *P. canaliculata* showed a more stable redox status between the two tidal periods in all sampling dates, while *F. serratus*, the species living at the lowest level of the intertidal, presented higher oxidative damage in the low tide period. Overall, the physiological responses analyzed were species-specific, demonstrating that low intertidal organisms such as *F. serratus*, which currently live in a more protected environment, were more susceptible to low tide, suggesting that at least this species will probably be dramatically affected by the expected effects of climate change.

## Figures and Tables

**Figure 1 plants-10-01892-f001:**
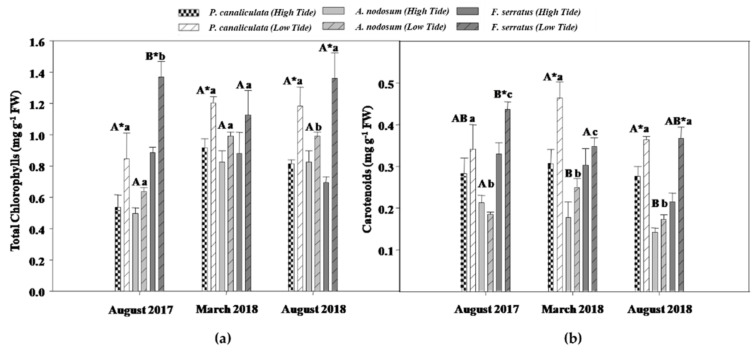
Photosynthetic pigments content of *P. canaliculata*, *A. nodosum*, and *F. serratus* at low and high tide conditions in August 2017, March and August 2018. (**a**) Total chlorophyll. (**b**) Carotenoids. Data presented are mean ± STDEV (*n* ≥ 3). * Above bars indicate significant statistical differences between tides at *p* ≤ 0.05. Different uppercase letters indicate significant statistical differences between species at high tide at *p* ≤ 0.05. Different lowercase letters indicate significant statistical differences between species at low tide at *p* ≤ 0.05.

**Figure 2 plants-10-01892-f002:**
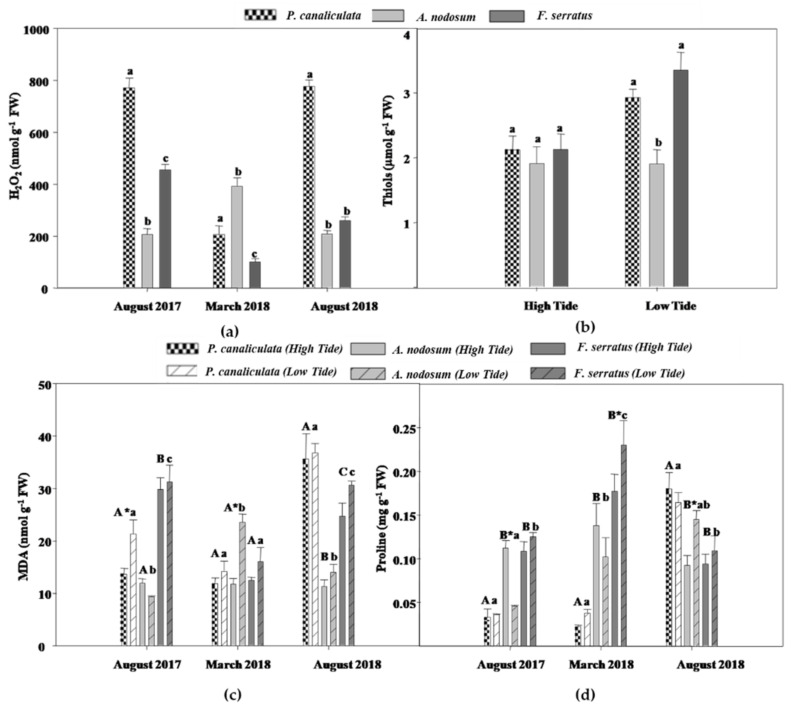
Oxidative stress indicators of *P. canaliculata, A. nodosum*, and *F. serratus* at low and high tide conditions in August 2017, March and August 2018. (**a**) H_2_O_2_. (**b**) Thiols. (**c**) MDA. (**d**) Proline. Data presented are mean ± STDEV (*n* ≥ 3). * Above bars indicate significant statistical differences between tides at *p* ≤ 0.05. Different uppercase letters indicate significant statistical differences between species at high tide at *p* ≤ 0.05. Different lowercase letters indicate significant statistical differences between species at low tide at *p* ≤ 0.05.

**Table 1 plants-10-01892-t001:** Information about water temperature and atmospheric temperature, UV index, and UVB irradiance during low and high tides of all sampling dates.

Sampling Date	Low Tide				High Tide			
	Water Temperature (°C)	Atmospheric Temperature (°C)	UV Index	UVB Irradiance (µW cm^−2^)	Water Temperature (°C)	Atmospheric Temperature (°C)	UV Index	UVB Irradiance (µW cm^−2^)
17-Aug	17.1	22	7	132.1	16.8	24	9	169.9
18-Mar	12.7	9	4	75.5	12.5	9	4	75.5
18-Aug	14.1	24	8	151	15.3	21	8	151

## Data Availability

The data presented in this study are available in this manuscript.

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
