# Peer review of "Fucoid Macroalgae Have Distinct Physiological Mechanisms to Face Emersion and Submersion Periods in Their Southern Limit of Distribution"

_plants, 2021, doi:10.3390/plants10091892_

Round 1

Reviewer 1 Report

see attached

Author Response

Reviewer’s comments (C) and author’s answers (A).

(A)

Thank you very much for all your suggestions. Your input was thoroughly considered and addressed in the revised manuscript, in the form of tracked changes. Also, we present point-by-point answers to your queries.

(C)

 Reviewer 1

I have reviewed manuscript plants-1332356 by Martins et al. The manuscript reports
measurements of several indicators of stress of three species of fucoid macroalgae due to
exposure during low tide. The algae, P. canaliculata, A. nodosum and F. serratus, were chosen because their habitat is relatively high, medium, and low in the intertidal zone and hence they are accustomed to being exposed for relatively long, medium, and short periods of time during low tide. I have several concerns with the revised version of this manuscript. First, the Materials and Methods section should provide enough information that someone else could repeat the experiments. The Materials and Methods section of this manuscript does not provide information that would be sufficient for someone else to repeat the experiments. The authors say that “The proportion of time that each species stayed exposed was the same in all dates” (lines 312–313). If I were going to try to repeat the experiments, I would obviously need to know what proportions and what time. What I would need to know is how much time, probably rounded to the nearest minute, each species was exposed in each experiment. There is an old adage: the dose makes the poison. In this case, the dose is directly proportional to the duration of exposure, but the authors never say what the duration of exposure of each species was. Without that information, it would be impossible for someone else to repeat the experiments. A second point concerns the depth of water below which the algae were submerged when they were submerged. Water is by no means opaque to ultraviolet radiation. See, for example, Fleischmann, E. M. 1989. The measurement and penetration of ultraviolet radiation into tropical marine water. Limnology & Oceanography 34(8): 1623–1629. Without knowing under how much water the algae were submerged, it is impossible to say how different the doses of ultraviolet radiation were when the algae were submerged and when they were exposed. Exposure to air (desiccation) is an either-or situation: either the algae were exposed to air or they were not. Exposure to ultraviolet radiation is different because water is not opaque to ultraviolet radiation. Finally, I think there is an unwarranted extrapolation at the end of the manuscript. F. serratus is the algal species is normally submerged more than the other species, and it was more sensitive to the effects of low tide in this study. However, the authors make the unwarranted extrapolation that (lines 390–391), “these species [meaning all species with habitats similar to that of F. serratus] will be dramatically affected by the expected effects of climate change.” Extrapolation of results on a single species to all species that live in similar habitats is unwarranted.

(A)

First of all, thank you for raising all those questions. All of them were considered and important to improve the quality of our manuscript. Your concerns are explained below:

  • Since our main goal was to assess differences between species living in distinct intertidal levels, canaliculata was the first species being exposed at low tide and thus, the first species being collected, followed by A. nodosum and then by F. serratus. In this sense, sampling at low tide was conducted immediately before the water covered the species to achieve the maximum exposure period for each studied macroalgae species. Thus, and considering that a tidal cycle has 12 h, the recorded emersion period of P. canaliculata was 10 h, followed by A. nodosum (8 h) and then by F. serratus (4 h). This information was added to our M&M section, in lines 334-337.
  • Concerning the issue regarding UV exposure, we think that is important to mention that the amount of UV that reaches macroalgae species when they are submerged is quite lower than that when those species are completely exposed. Nonetheless, and considering that the depth of water that below which species were submerged has a great influence on the quantity of UV radiation, at the sampling moment during high tide (which corresponded to the period that macroalgae were completely submerged), the depth of water reached 0.5-0.75 m in canaliculata; 1.0-1.30 m A. nodosum; 2.0-3 m in F. serratus. This information was added to our M&M section, in lines 329-331.
  • Moreover, considering that macroalgae have distinct responses to face unfavourable conditions, we cannot extrapolate the consequences of CC for all low tide organisms. Therefore, and in line with your rationale, that sentence was altered.
  • Lastly, as the reviewer suggests, we are well aware that field studies are way more complex than those of laboratory, in which all variables can be controlled and manipulated. Thus, and although we have tried to record several parameters to provide a robust workflow and reproducibility of our data, we know that many variables could be also accounting for some variation. For that reason, we have collected five experimental replicates in each sampling date to achieve a representative sampling of each studied species.

Reviewer 2 Report

The authors took into account the main part of my previous comments in the resubmitted manuscript.  I appreciate it.

However, I did not get answers to some my questions as well as I can see inaccuracies in the present text. In my opinion, authors of the articles have to work on the final text of a manuscript with an increased attention.

Why the authors did not collect samples in the second March? This would make it possible to compare two full-year cycles. 

The authors refer to Lichtenthaler for the determination of total Chl and Car. This method allows also to determine separately Chl a and Chl b that make it possible to evaluate the state of an antenna of photosystems as a ratio of Chl a to Chl b. Try to calculate it from the absorption spectra that you have.

I think that the determination of Car composition in the studied samples is a very interesting and important moment of the study. If the authors have the possibility to do it with HPLC, I would ask them to add such data to this manuscript.

Line 27. PelvetiaP.

Line 116 and below in the text. “…,being 0.6- and 1-fold higher than…”. I can only propose that 1.6 and 2-fold are more correct here, or the authors mean about 60 and 100%? Can the authors comment these moments?

Lines 125 and 128. Does December mean August?

The legend of Fig. 2b have to be corrected: P. canaliculata (Low Tide) → P. canaliculata (High Tide) and vice versa. In addition, I would recommend moving up the paragraph about Thiols to it follows immediately after the ROS description (Line 170 to 152). The panel c has to be b, respectively. Delete “dates 1, 2 and 3” in the Fig.2 description (Lines 153-154).

Correct the order of individuals in the beginning of the Material (Line 304).

Author Response

Please see the atachment

Round 2

Reviewer 1 Report

see attached

Author Response

First of all, thank you for the information presented in your revision. Indeed, and after reading your point of view, we understand that a different method to measure UV radiation should have been employed, due to its importance in this type of studies. Unfortunately, we did not possess the necessary equipment for such measurements. However, it is a crucial aspect that we will certainly take into consideration in future experiments in order to ensure a higher scientific soundness. Nonetheless, and although we did not have the exact values for the UV radiation, the UV index did not vary among tides and dates, so we believe that our main goal (to assess differences between species belonging to distinct positions at the intertidal towards two tidal regimes) was accomplished.

Reviewer 2 Report

I think that the manuscript can be accepted in the present view.

I want to pay attention of the authors to only some moments, which have no strong influence on the main text.

Line 118. Is it correct to talk about a 130% decrease of the Chl concentration? What is 100% in this case? If 100% is a value from F. serratus (1.4), then a value from A. nodosum (0.6) is decreased to ~40% or by ~60%, but not by 130%. If 100% is a value from A. nodosum, then a value from F. serratus is higher by ~230%.

Thus the authors should check the correctness of all values given in % throughout the entire text.

The legend of Fig. 2c still has to be corrected: P. canaliculata (Low Tide) → P. canaliculata (High Tide) and vice versa.

Author Response

This manuscript is a resubmission of an earlier submission. The following is a list of the peer review reports and author responses from that submission.

Round 1

Reviewer 1 Report

In order to deepen the physiological adaptation it would be interesting to investigate the activities of ROS scavenging enzymes.

Moreover it would be intriguing if it is not yet investigated to control adaptations at varying temperature conditions

Reviewer 2 Report

see attached

Reviewer 3 Report

A brief summary

The present work describes differences in some biochemical parameters observed by the authors in three species of macroalgae obtained in different seasons of the year from an intertidal area. The manuscript is written in a good manner, but I have some points to comment listed below.

Broad comments

The authors should check all moments in the text where they write the names of the algae because the full name is necessary only once.  For example, in Line 26 Pelvetia canaliculata P. canaliculata. The same in lines 70, 73, 113, 121, 162, and etc.

The abbreviation CC (Climate change) is not needed in the manuscript in my opinion.

The authors collected samples from three different dates, two of them in August and one in March. Why the authors did not collect samples in the second Mach? This would make it possible to compare two full-year cycles.  

I am not sure that the representation of Tables with ANOVA calculations is necessary for the main text. Could the authors think about moving them to supplementary materials?  In addition, can the authors explain why Data (Da) in Table 1 do not have significant differences (Line109) because p is 0.0026? The same in Table 2 for Da, and etc.

In Fig 1 (and other figs), I would recommend writing algae names as P. canaliculata but not as Pelvetia in legends. Moreover, the order of the data for each alga in figures should be the same as the authors introduce before in the text and as they writhe in figures captions: (1) A. nodosum, (2) F. serratus and (3) P. canaliculata (the same is in the Abstract (Line 21) and Introduction (Line 70)). In figures panels the order is P. canaliculata, A. nodosum, F. serratus. Correct it in all figures.

Change the design of all figures to the label of the Y-axis was on the left side. Now it looks, that all panels have data from the two scales. This is incorrect. I would recommend changing labels “Date” on the real date (August 2017, March and August 2018) in Figs.

Figures 2 should be after the first meaning of it in the text (Line 147), i.e. in page 5.

Line 176. Pro is Proline? I think that such reduction is incorrect.

The authors refer to Lichtenthaler for the determination of total Chl and Car. This method allows also to determine separately Chl a and Chl b that make it possible to evaluate the state of an antenna of photosystems as a ratio of Chl a to Chl b. Try to calculate it from the absorption spectra that you have.

Table 3. “Concerning total thiols, significant differences for the Species x Tide interaction (Sp 165 x Ti; Table 3) were found.” (Line 165). What about Sp (p=0.003), Ti (p=0.019), Da (p=0)? Explain.

I think that the determination of Car composition in the studied samples is a very interesting and important moment of the study. If the authors have the possibility to do it with HPLC, I would ask them to add such data to this manuscript.

Line 220. Pigments can not be denatured. Correct it.

Lines 229-232. “Also, this species is known to promptly activate non-photochemical quenching mechanisms, such as the xanthophyll cycle, in order to reduce excess energy, converting violaxanthin into antheraxanthin and zeaxanthin, an important response mechanism to high light intensity [13].” The studies of NPQ in such microalgae is probably a very important area of research now because we know almost nothing about it as compare to higher plants and green algae. In addition, NPQ is one of the rapid mechanisms allowing photosynthetic apparatus to adapt to changed conditions in minutes. I would recommend the authors pay attention to this problem in their future works.

Table 5. I would recommend to the authors to change WT → water, and AT → atmosphere because WT usually means wild-type of organisms.

Is SN supernatant? Do not reduce it.